# Effects of Heated Drinking Water during the Cold Season on Serum Biochemistry, Ruminal Fermentation, Bacterial Community, and Metabolome of Beef Cattle

**DOI:** 10.3390/metabo13090995

**Published:** 2023-09-06

**Authors:** Tengfei He, Guang Yi, Xilin Wang, Yan Sun, Jiangong Li, Zhenlong Wu, Yao Guo, Fang Sun, Zhaohui Chen

**Affiliations:** 1College of Animal Science and Technology, China Agricultural University, Beijing 100193, China; hetengfei@cau.edu.cn (T.H.); sy20203040712@cau.edu.cn (G.Y.); sy20213040719@cau.edu.cn (X.W.); sunyan93@cau.edu.cn (Y.S.); jli153@cau.edu.cn (J.L.); wuzhenlong@cau.edu.cn (Z.W.); guoyao@cau.edu.cn (Y.G.); 2State Key Laboratory of Animal Nutrition and Feeding, Beijing 100193, China; 3Institute of Animal Huabandry, Heilongjiang Academy of Agricultural Sciences, Harbin 150086, China; hff060078@gmail.com

**Keywords:** water temperature, cold season, rumen function, bacterial community, metabolome, beef cattle

## Abstract

This study explored the effects of drinking heated water in the cold seasons on the serum metabolism, rumen microbial fermentation, and metabolome of beef cattle. Twelve fattening cattle (642 ± 14.6 kg) aged 21 to 22 months were randomly and equally divided into two groups based on body weight: one receiving room-temperature water (RTW; average 4.39 ± 2.55 °C) and the other heated water (HW; average 26.3 ± 1.70 °C). The HW group displayed a significant decrease in serum glucose (*p* < 0.01) and non-esterified fatty acid (*p* < 0.01), but increases in insulin (*p* = 0.04) and high-density lipoprotein (*p* = 0.03). The rumen fermentation parameters of the HW group showed substantial elevations in acetate (*p* = 0.04), propionate (*p* < 0.01), isobutyrate (*p* = 0.02), and total volatile fatty acids (*p* < 0.01). Distinct bacterial composition differences were found between RTW and HW at the operational taxonomic unit (OTU) level (R = 0.20, *p* = 0.01). Compared to RTW, the HW mainly had a higher relative abundance of *Firmicutes* (*p* = 0.07) at the phylum level and had a lower abundance of *Prevotella* (*p* < 0.01), *norank_f_p-215-o5* (*p* = 0.03), and a higher abundance of *NK4A214_group* (*p* = 0.01) and *Lachnospiraceae_NK3A20_group* (*p* = 0.05) at the genus level. In addition, *NK4A214_group* and *Lachnospiraceae*_*NK3A20*_*group* were significantly positively correlated with the rumen propionate and isovalerate (r > 0.63, *p* < 0.05). *Prevotella* was negatively correlated with rumen propionate and total volatile fatty acids (r = −0.61, *p* < 0.05). In terms of the main differential metabolites, compared to the RTW group, the expression of Cynaroside A, N-acetyl-L-glutamic acid, N-acetyl-L-glutamate-5-semialdehyde, and Pantothenic acid was significantly upregulated in HW. The differentially regulated metabolic pathways were primarily enriched in nitrogen metabolism, arginine biosynthesis, and linoleic acid metabolism. *Prevotella* was significantly positively correlated with suberic acid and [6]-Gingerdiol 3,5-diacetate (r > 0.59, *p* < 0.05) and was negatively correlated with Pantothenic acid and isoleucyl-aspartate (r < −0.65, *p* < 0.05). *NK4A214_group* was positively correlated with L-Methionine and glycylproline (r > 0.57, *p* < 0.05). Overall, our research demonstrates the important relationship between drinking water temperature and metabolic and physiological responses in beef cattle. Heating drinking water during cold seasons plays a pivotal role in modulating internal energy processes. These findings underscore the potential benefits of using heated water as a strategic approach to optimize energy utilization in beef cattle during the cold seasons.

## 1. Introduction

The rumen, fundamental to ruminant nutrition, digestion, absorption, and energy utilization, plays a pivotal role in beef cattle health and growth performance [1]. A thriving bacterial community resides within the rumen, executing complex metabolic processes, which encompass both feed fermentation and the utilization of resultant metabolites for bacterial proliferation [2,3]. The internal environment of the rumen is a key factor in maintaining optimal bacterial activity, notably the rumen temperature [4,5]. External influences, including ambient and water temperatures, could substantially modify the rumen temperature [6,7]. Ruminants continuously interact thermally with their environment [8]. In colder conditions, these interactions could decrease the body temperature and undermine performance, impairing the immunity and healthy status of cattle [9,10].

In response to the detrimental effects of cold conditions, supplying heated drinking water has been proposed as a proactive measure. Water is a pivotal nutritional component, and the temperature of consumed water profoundly impacts rumen temperature variations under cold conditions. For instance, Petersen et al. [11] demonstrated that the percentage of cows exhibiting a rumen temperature below 38 °C was significantly reduced in cows provided with warm water at approximately 31.1 °C, as opposed to cows provided with room-temperature water at around 8.2 °C. Additionally, the consumption of room-temperature water or warm water resulted in the lowest recorded rumen temperature of 31.6 °C or 34.5 °C, respectively. A rumen temperature drop could inhibit microbial functions [12] and diminish microbial adhesion to fibrous matter [13]. Moreover, empirical studies indicate a high correlation between rumen temperature and core body temperature [14], and a reduced core temperature could further induce oxidative stress and immune function degradation [15,16], compromising ruminant health and welfare. Thus, supplying heated water in colder climates emerges as an efficacious strategy, demonstrated by the notable enhancement of fattening cattle’s growth, antioxidant properties, and stress resilience [17].

Despite some research underlining the significance of water temperature on ruminant performance, the underlying mechanisms remain unclear. Cutting-edge techniques, including 16S rRNA gene sequencing and liquid chromatography–mass spectrometry analysis, have paved the way for obtaining a holistic insight into the dynamics between rumen microbiota and metabolites. This study was conducted to probe into the repercussions of water temperature during colder seasons on the serum biochemistry, rumen microbiota, and metabolites of beef cattle. The aim of this study is to gain insight into the role of heated drinking water in promoting health and rumen fermentation in beef cattle, thereby informing and enriching livestock feeding management strategies during the cold season.

## 2. Materials and Methods

The study received authorization from China Agricultural University’s Animal Welfare and Ethics Committee (permit number AW71012022-1-2) and was conducted at the Lianwang Beef Cattle Research Facility in Henan, China.

### 2.1. Animals, Design, and Management

A total of 12 fattening cattle, with an average body weight (mean ± standard deviation) of 642 ± 14.6 kg, aged 21 to 22 months, were selected and randomly and equally divided into two groups based on body weight: room-temperature water (average drinking water temperature of 4.39 ± 2.55 °C) and heated water (average drinking water temperature of 26.3 ± 1.70 °C), named RTW and HW, respectively. Each treatment consisted of 6 fattening cattle, and the experimental phase spanned 60 days. Each beef cattle in the HW group were provided with an automatic electric heating water tank (Kangkaijie Agricultural Technology Co., Ltd., Beijing, China) fitted with a temperature sensor to ensure the drinking water remained at the necessary temperature for the experiment. All feeding protocols, with the sole exception of drinking water temperature, were maintained identically across both treatments. The diet’s composition and nutrient content, delineated in Table 1, complied with the guidelines established by the Committee on Nutritional Requirements [18]. The diet was dispensed as a total mixed ration at two fixed intervals daily, at 7:00 a.m. and 3:00 p.m., with the residual feed from the preceding day being cleared at 6:30 a.m. to facilitate ad libitum feeding. Drinking water temperature was ascertained and documented at four distinct time points each day (8:00, 12:00, 16:00, and 18:00) using a Testo 635 device (Testo International Trade Co., Ltd., Shanghai, China). Furthermore, the ambient temperature and relative humidity were perpetually monitored at half-hour intervals, situated 1.7 m above ground level, via a specialized temperature and humidity recorder (Apresys 179A-TH, Apresys Optoelectronics Co., Ltd., Shanghai, China). Throughout the duration of the experiment, the average (±SD) temperature, relative humidity, and THI were 2.15 ± 6.05 °C, 72.6 ± 3.85%, and 39.1 ± 9.46, respectively. The highest and lowest ambient temperatures recorded were 17.1 °C and −12.4 °C, respectively.

### 2.2. Sample Collecion

On the 61st day, approximately 5 mL of blood was drawn via the tail vein using heparinized tubes at 6:00 a.m. from each cattle. Upon collection, the blood samples were subjected to immediate centrifugation at a force of 3000× *g* for a duration of 15 min at a controlled temperature of 4 °C and were subsequently stored in a freezer at −80 °C for future examination. Ruminal fluid samples were obtained at 9:00 a.m. on the same day by employing aspiration via an esophagogastric tube. During rumen fluid extraction, the initial 200 mL was discarded. The remaining fluid was then filtered through four layers of sterile gauze. Subsequently, this filtered rumen fluid was aliquoted into three 2.0 mL sterile storage tubes (NEST Biotech Co., Ltd., Jiangsu, China) and was cryopreserved in liquid nitrogen for future analysis.

### 2.3. Chemical Analysis

#### 2.3.1. Serum Index

Biochemical indicators in the serum, including insulin (INS), non-esterified fatty acids (NEFAs), total cholesterol (TC), triglycerides (TGs), high-density lipoproteins (HDLs), low-density lipoproteins (LDLs), total proteins (TPs), glucose concentrations (Glu), and blood urea nitrogen (BUN), were measured using a fully automated biochemical analyzer (Hitachi 7020, Hitachi Co., Tokyo, Japan). All assessments followed the guidelines set out in the accompanying kit (Beijing Jiuqiang Biotechnology Co., Ltd., Beijing, China). Furthermore, the serum levels of immunoglobulin A (IgA), immunoglobulin M (IgM), and immunoglobulin G (IgG) were determined via enzyme-linked immunosorbent assay techniques, adhering to the protocols provided in the respective kit (Jiangsu Enzyme Industrial Co., Ltd., Jiangsu, China).

#### 2.3.2. Rumen Fermentation Parameters

The rumen fluid’s pH was swiftly determined using a digital pH meter (PHS-3C; Shanghai Yueping Scientific Instrument Co., Ltd., Shanghai, China). Ammonia nitrogen levels were measured following the protocol outlined by He et al. [17], with the aid of a spectrophotometer (UV-1700, Shimadzu Corporation, Kyoto, Japan). Concurrently, the concentration of volatile fatty acids (VFAs) in the rumen fluid was precisely analyzed via a high-performance gas chromatograph (GC-8600; Beifen Tianpu Instrument Technology Co., Ltd., Beijing, China).

### 2.4. Rumen Microbiota Analysis

DNA from the microbial populace was isolated using the E.Z.N.A.^®^ Soil DNA Kit (Omega Bio-tek, Norcross, GA, USA). The integrity of the isolated DNA was assessed using 1% agarose gel electrophoresis. The V3-V4 variable regions of the 16S rRNA gene were then amplified via PCR using the primers 338F and 806R. The amplification process began with a 95 °C denaturation step for 3 min. This was followed by 27 cycles: each cycle involved denaturation at 95 °C for 30 s, annealing at 55 °C for 30 s, and elongation at 72 °C for 30 s. The final elongation step lasted for 10 min at 72 °C. An ABI GeneAmp^®^ 9700 thermal cycler (Applied Biosystems, Waltham, MA, USA) was utilized for the PCR.

PCR yields from the same samples were pooled, cleaned, and quantified precisely. The NEXTFLEX Rapid DNA-Seq Kit was employed to construct the sequencing library, involving steps like adapter addition, the removal of self-joined fragments, further PCR amplification, and bead-based purification. Sequencing was carried out on the Illumina MiSeq PE300/NovaSeq PE250 system by Shanghai Meiji Biomedical Technology Co., Ltd. (Shanghai, China) Sequence cleaning and merging were managed using the fastp [19] and FLASH [20] software tools. The resultant sequences were clustered into operational taxonomic units (OTUs) at a similarity threshold of 97% [21,22], with any chimeric sequences duly removed. The RDP classifier [23] (http://rdp.cme.msu.edu/, version 2.2, accessed on 7 May 2023) was employed, cross-referencing the Silva 16S rRNA database (v138) with a comparison threshold set at 70%.

### 2.5. Non-Targeted Metabolomics Analysis

We followed the protocol detailed by Ogunade et al. [24] for handling the rumen fluid specimens. Specifically, we mixed 500 µL of each rumen fluid sample with 2 mL of a 1:1 methanol–water solution, followed by vortexing for 2 min. This mixture was then centrifuged at 15,000× *g* for 10 min at 4 °C. The extracted supernatant underwent dehydration in a vacuum concentrator, and the residual content was re-dissolved in a 200 µL solution of methanol and water in equal parts. The analysis was facilitated through the UltiMate™ 3000 UPLC system (Thermo Fischer Scientific, Waltham, MA, USA), integrated with an autosampler and an Orbitrap-Velos MS. We utilized an Agilent Extend C-18 column (3.0 × 150 mm, 3.5 µm; Agilent, Santa Clara, CA, USA) for chromatographic separation, operating at 45 °C, while preserving the sample handler at 4 °C. We defined UPLC mobile phases as (A) 0.1% formic acid in water and (B) 0.1% formic acid in acetonitrile. The system was set to an Injection volume of 5 µL and a flow rate of 0.5 mL/min. MS analysis was performed in both ionization polarities with a capillary voltage set to 3.5 kV. We kept the capillary and source temperatures at 350 °C and set the sheath and auxiliary gas flow rates to 40 L/h and 10 L/h, respectively. For system consistency and precision, pooled quality control (QC) samples from each rumen fluid were integrated after processing every four samples.

Initial data transformation was achieved with the Reifycs ABF Converter (http://www.reifycs.com/AbfConverter/index.html, accessed on 7 May 2023). Subsequent data refinement was facilitated by the MS-DIAL software (version 2.84) [25].

### 2.6. Statistical Analysis

Analyses of serum data and parameters from rumen fermentation were executed using the unpaired Student’s *t*-test approach. To discern variations in the rumen liquid microbiota abundance, the LEfSe method was employed, leveraging the Kruskal–Wallis rank sum evaluation. Effect sizes in the dataset were signified using LDA scores, setting a benchmark at ≥2.50. To undergo multivariate/univariate evaluation, metabolites discerned in both positive- and negative-ion modes were amalgamated and processed through MetaboAnalyst 4.0 [26]. Based on the outcomes of the *t*-test and fluctuations in the peak intensity, differential metabolites were isolated and verified. Utilizing the Bos taurus pathway repository, pathway examination was facilitated through MetaboAnalyst 4.0 software. A threshold of the *p* value ≤ 0.05 was set to determine significant disparities between the study groups. For *p* values ranging between 0.05 and 0.10, the observed difference was interpreted as a trend.

## 3. Results

### 3.1. Serum Parameters

In comparison to the RTW group, the HW group exhibited a significant reduction in serum Glu (*p* = 0.01) and NEFAs (*p* < 0.01), concomitant with a significant elevation in serum INS (*p* = 0.04) and HDLs (*p* = 0.03). Moreover, a trending increase in serum IgA (*p* = 0.06) and TGs (*p* = 0.10) was observed in the HW group relative to RTW (Figure 1).

### 3.2. Rumen Fermentation Parameters

As shown in Figure 2, compared to RTW, the concentrations of acetate (*p* = 0.04), propionate (*p* < 0.01), isobutyrate (*p* = 0.02), and T-VFAs (*p* < 0.01) in the rumen fluid of HW significantly increased, while the ratio of acetate to propionate showed a decreasing trend (*p* = 0.06) of HW.

### 3.3. Bacterial Sequencing

An analysis was conducted on rumen fluid samples from 12 beef cattle, resulting in a total of 700,224 refined sequences, each averaging 416 bp in length (Table 2). By implementing a random subsampling strategy based on the fewest sequences in a sample, we identified 2594 OTUs. These OTUs were subsequently categorized into 32 phyla, 81 classes, 180 orders, 326 families, 663 genera, and 1141 species after aligning with the Silva database.

### 3.4. Bacterial α-Diversity

In terms of alpha diversity, no significant differences were observed between the RTW and HW groups in the indices of Sob, Shannon, Simpson, Ace, Chao, and Coverage (Figure 3).

### 3.5. Bacterial Composition and β-Diversity Analysis

Venn analysis revealed the presence of 1441 shared operational taxonomic units (OTUs), along with 680 unique OTUs in the RTW group and 473 unique OTUs in the HW group (Figure 4A). Principal coordinate analysis (PCoA) at the OTUs level demonstrated significant differences between the RTW and HW groups (PCoA: R = 0.202, *p* = 0.014) (Figure 4B). The microbial composition was visualized through bar graphs at both the phylum (Figure 4C) and genus (Figure 4D) levels. The top three microbial phyla within the RTW and HW groups were identified as Firmicutes, Bacteroidetes, and Actinobacteria. At the genus level, the predominant microbial taxa within both RTW and HW groups were *Prevotella*, *Rikenellaceae_RC9_gut_group*, *norank_f_Muribaculaceae*, *NK4A214_group*, and *Christensenellaceae_R-7_group*.

### 3.6. Bacterial Components Differences Analysis

The primary differences in bacterial composition between the RTW and HW groups are illustrated in Figure 5. Compared to the RTW group, the HW group exhibited a higher relative abundance of the phylum Firmicutes (*p* = 0.07) (Figure 5A) and a lower relative abundance at the genus level of *Prevotella* (*p* < 0.01) and *norank_f_p-215-o5* (*p* = 0.03), along with an increased abundance of *NK4A214_group* (*p* = 0.01) and *Lachnospiraceae_NK3A20_group* (*p* = 0.05) (Figure 5B). Furthermore, the LEfSe analysis (Figure 5D) disclosed additional distinct bacterial taxa. Relative to the RTW group, the HW group demonstrated a significant decrease in the relative abundance of *Succinivibrionaceae* and *Alloprevotella*, and a concomitant significant increase in the relative abundance of *Oscillospiraceae* and *Defluviitaleaceae* (LDA > 2.50, *p* < 0.05).

### 3.7. Examination of Correlation among Dominant 15 Bacterial Genera and Fermentation Parameters

Based on the Spearman correlation assessment (Figure 6), the genera *Acetitomaculum*, *NK4A214*_*group*, and *Lachnospiraceae*_*NK3A20*_*group* exhibited a strong positive association with rumen propionate and isovalerate (r > 0.63, *p* < 0.05), which also displayed an inverse relationship with the acetate-to-propionate ratio (r < −0.63, *p* < 0.05). Notably, acetate and T-VFAs showed a positive association with bacterial groups *norank*_*f*__*Muribaculaceae* and *Christensenellaceae*_*R-7_group* (r > 0.58, *p* < 0.05). Meanwhile, *Prevotella* and *norank_f__p-251-o5* demonstrated a negative association with both rumen propionate and T-VFAs (r < −0.61, *p* < 0.05).

### 3.8. Comparative Analysis of Rumen Metabolites

Multiple statistical evaluation methods, including principal component analysis (PCA) and discriminant analysis via partial least squares (PLSs-DA), were employed to investigate the underlying associations between metabolomics and biological characteristics. Analyses were conducted under both positive-ionization mode (Figure 7A) and negative-ionization mode (Figure 7B) for PCA and PLS-DA. The outcomes delineated a conspicuous separation between the rumen fluid samples of the RTW and HW groups, underscoring the existence of significant differences in rumen fluid metabolism between the two groups.

### 3.9. Rumen Metabolites Components Differences Analysis

A total of 202 named differential metabolites were identified in rumen fluid samples from the RTW and HW groups using LC-MS analysis and selection criteria of VIP > 1.00 and *p* < 0.05. Among these metabolites, 85 were identified in positive-ionization mode (Figure 8A), and 117 were identified in negative-ionization mode (Figure 8B). Among the identified metabolites, 133 showed significantly decreased levels, while 69 displayed the opposite trend. These differential metabolites primarily belonged to the classes of organic acids and derivatives, lipids and lipid-like molecules, organic oxygen compounds, phenylpropanoids, and related compounds. Table 3 presents the major differential metabolites between the two treatment groups. Compared to the RTW group, the expression of 2-Formaminobenzoylacetate, 4-Methyl-1-phenyl-2-pentanol, 5,6-Dihydroxyprostaglandin F1a, 6-Amino-9H-purine-9-propanoic acid, 7-Ketodeoxycholic acid, Carboxyibuprofen, Glutamic acid, N-Vinyl-2-pyrrolidone, Nigellic acid, and Tyrosine was significantly downregulated in the HW group, whereas the expression of 1-Aminocyclopropane-1-carboxylic acid, 3-Methyloxindole, 6-Hydroxy-1H-indole-3-acetamide, Cynaroside A, D-Glucurone, D-Glucuronic acid, D-Pipecolic acid, Formiminoglutamic acid, Formylisoglutamine, Hydantoin-5-propionic acid, Indole-3-propionic acid, Linoelaidic acid, Melibiitol, N-acetyl-L-glutamic acid, N-acetyl-L-glutamate-5-semialdehyde, Orthothymotinic acid, and Pantothenic acid was significantly upregulated. Enrichment analysis of the KEGG pathway (Figure 9) indicated that the differentially regulated metabolic pathways were primarily enriched in nitrogen metabolism, two-component system, neuroactive ligand–receptor interaction, proximal tubule bicarbonate reclamation, toluene degradation, D-glutamine and D-glutamate metabolism, synaptic vesicle cycle, lysine biosynthesis, GABAergic synapse, glutamatergic synapse, cysteine and methionine metabolism, phenylalanine metabolism, aminoacyl-tRNA biosynthesis, central carbon metabolism in cancer, biosynthesis of plant secondary metabolites, arginine biosynthesis, linoleic acid metabolism, alanine, aspartate, and glutamate metabolism, protein digestion and absorption, and histidine metabolism.

### 3.10. Spearman Correlation Analysis of the Top 50 Bacteria Genus with Metabolites in Rumen

According to the Spearman correlation analysis (Figure 10), *Prevotella* was significantly positively correlated with 2-Formaminobenzoylacetate, 5,6-Dihydroxyprostaglandin_F1a, valerenolic acid, 2,4-octadienal, suberic acid, and [6]-Gingerdiol 3,5-diacetate (r > 0.59, *p* < 0.05) and was negatively correlated with pantothenic acid, N-Acetyl-L-glutamic acid, L-(−)-Tyrosine, L-4-Hydroxyglutamate semialdehyde, L-Methionine S-oxide, (R)-(+)-2-Pyrrolidone-5-carboxylic acid, (S)-2-Azetidinecarboxylic acid, formylisoglutamine, and isoleucyl-aspartate (r < −0.65, *p* < 0.05). *NK4A214_group* was positively correlated with L-4-Hydroxyglutamate semialdehyde, L-Methionine S-oxide, (R)-(+)-2-Pyrrolidone-5-carboxylic acid, formylisoglutamine, and glycylproline (r > 0.57, *p* < 0.05). *Lachnospiraceae_NK3A20_group* was positively correlated with L-Methionine S-oxide, 9(S)-HpODE, orthothymotinic acid, and isoleucyl-glutamate (r > 0.57, *p* < 0.05).

## 4. Discussion

The findings of our study elucidated the multifaceted effects of HW on the serum biochemistry of beef cattle during the cold season, weaving together aspects of energy metabolism, lipid metabolism, glucose regulation, and immune function. The observed increase in NEFAs in the RTW group aligned with previous findings that NEFA concentrations typically increased under cold conditions to support energy demands through lipolysis, leading to heat generation for maintaining body temperature [27,28]. This increase in NEFAs, coupled with the concurrent increase in Glu, might suggest an enhanced need for thermogenesis in the RTW group, compared to HW. Furthermore, considering the role of NEFAs in activating inflammatory signaling pathways and influencing insulin resistance [29], these changes might also signal an enhancement in metabolic health and a decrease in inflammation. Alongside these changes, the significant increase in HDLs in the HW group could be indicative of improved cardiovascular health through altered lipoprotein metabolism, reflecting findings from previous research on ambient temperature’s effect on lipoproteins [30]. Additionally, the simultaneous increase in serum INS of the HW group could be interpreted by previous studies that showed that low temperatures reduced insulin secretion [31], suggesting that HW might mitigate cold-induced alterations in glucose regulation. Noteworthy as well were the trends towards increased IgA and TGs in the HW group, which, though not statistically significant, could hint at underlying changes in immune function and lipid profiles [28], warranting further investigation.

Increased concentrations of acetate, propionate, isobutyrate, and T-VFAs were observed in the HW group, which partly agreed with recent in vivo and in vitro reports, demonstrating that drinking cold water or periodically lowering the in vitro incubation temperature led to decreased propionate and T-VFA concentrations in rumen or broth liquid [17,32]. Given that propionate served as a crucial substrate for gluconeogenesis and played a vital role in sustaining energy homeostasis in vivo [33], the observed reduction in ruminal propionate in the RTW group could correspond to an augmented need for propionate uptake to facilitate gluconeogenesis during cold conditions. This interpretation was further supported by the simultaneous increase in serum Glu and NEFAs and decrease in INS observed in the RTW group, reflecting the enhanced energy demand imposed by cold drinking water in winter, which in turn might accelerate the glucose and lipid metabolism in beef cattle [28]. Moreover, the utilization of cold water in winter could substantially lower the rumen temperature of ruminants [34], and the decline in ruminal VFA production in RTW beef cattle could be ascribed to diminished microbial adhesion to fibers and a concurrent reduction in the concentration of digestive enzymes at lower temperatures [13]. The observed decrease in ruminal VFAs in RTW beef cattle might signal an adaptive response to cold stimulus, underscoring the potential utility of HW as a strategic approach to mitigate such stimulus and optimize energy utilization within the organism.

Though alpha diversity did not reveal significant differences between RTW and HW groups, a significant distinction was observed at the OTU level through PCoA. This finding was congruent with previous studies by He et al. [17] and Duarte et al. [35], emphasizing that temperature exerted an obvious influence on microbial community structure. The relative abundance of Firmicutes was decreased in the RTW group, which mainly degraded the cellulose and were recognized as beneficial gut bacteria vital for herbivores’ health and growth [36,37], indicating that heating drinking water in winter is beneficial for beef cattle. Additionally, the RTW diminished *Prevotella*, *norank_f_p-215-o5*, and *Succinivibrionaceae*, consistent with the findings of He et al. [32] and Cui et al. [38]. The prominence of *Prevotella*, principally engaged in degrading non-structural carbohydrates and fostering propionate production [39], aligns with the hypothesis that cold water intake induced beef cattle’s augmented energy requirements during cooler periods. Furthermore, our correlation analysis revealed an interplay between microbiota and fermentation parameters. The genera *NK4A214_group* and *Lachnospiraceae_NK3A20_group* were significantly positively correlated with rumen propionate and isovalerate, which mirrored Liu et al. [40], where a higher abundance of *NK4A214_group* was found in yaks fed on highly concentrated diets, emphasizing its role in refractory starch digestion and thus providing a possible explanation for its positive correlation with propionate, butyrate, and isobutyrate. The *Lachnospiraceae_NK3A20_group*, identified for fermenting glucose into lactic acid [41], was abundant in the high-starch diet [42], further substantiating its positive correlation with propionate. Additionally, the relative abundance of *Succinivibrionaceae* increased in the RTW group, which could enhance digesting and utilizing substrates like starch, hemicellulose, pectin, and protein [43], and transforming succinate into propionate [44]. The dominance of *Prevotella* in the RTW group, coupled with increased *Succinivibrionaceae*, tied into existing research highlighting their contributions to rumen VFAs synthesis, which might enhance the generation of energy substrates for beef cattle under cold conditions and play an instrumental role in their absorption and utilization [45,46].

Our results indicated interactions between drinking water temperature, rumen microbial metabolism, and physiological outcomes of beef cattle. Both PCA and PLS-DA analysis modes showed a clear separation between the rumen fluid samples of the RTW and HW groups in either positive- or negative-ion mode, indicating significant differences in rumen fluid metabolism between the two groups. These differences were chiefly manifested in the categories of organic acids, lipids, and other pertinent compounds, suggesting that the metabolic alterations stemming from cold season drinking water temperature primarily pertain to internal energy processes [47]. A marked upregulation in linoleic acid metabolism substantiated the notion that, during the cold season, beef cattle intensively metabolized essential fatty acids, like linoleic acid, adapting to the nutritional and environmental challenges presented by low temperatures [39]. In the ruminal context, the conversion of glucose-1-phosphate, derived from feed degradation, to D-glucuronic acid via dehydrogenation and hydrolysis processes is essential [48]. Elevated D-glucuronic acid levels in the HW group suggested an augmentation of carbohydrate metabolic activity in beef cattle consuming heated water [42]. Pantothenic acid, a cornerstone of coenzyme A, played a pivotal role in the metabolism of fatty acids and ketone bodies within animals [49,50]. The diminished presence of pantothenic acid in the RTW group might signify a contraction in fatty acid and ketone metabolism. Furthermore, the metabolic profile in the rumen agreed with the observed serum parameters, including decreased Glu and NEFAs and elevated INS and HDLs in the HW group, indicating the physiological interplay mediated by metabolites like D-glucuronic acid and pantothenic acid in colder conditions.

This investigation also found positive correlations between genera like *Prevotella*, *NK4A214_group*, and *Lachnospiraceae_NK3A20_group* and amino acid metabolites such as glycylproline and isoleucyl-glutamate. Given that these bacterial genera predominantly engage in fiber and starch degradation, it was plausible to posit that bacterial carbohydrate breakdown might drive the observed shifts in amino acid metabolic routes [42]. In addition, we observed that N-acetyl-L-glutamate-5-semialdehyde and N-acetyl-L-glutamic acid exhibited pronounced upregulation in the HW group. These metabolites predominantly act as forerunners in the arginine biosynthesis pathway [51]. The literature indicate that rumen microbes have the aptitude to harness VFAs and other compounds as carbon sources, while using ammonia and related nitrogen compounds for nitrogen, facilitating de novo amino acid synthesis [52,53]. Within this study’s framework, augmented propionate and T-VFA levels in the HW group seemingly offer a richer substrate pool for rumen-based amino acid synthesis. This lends weight to the assertion that beef cattle in the HW group might exhibit an enhanced capacity for rumen amino acid metabolism relative to their RTW counterparts.

## 5. Conclusions

This study assessed the impact of drinking water temperature on the serum physiology, rumen microbial composition, and metabolism of beef cattle during the cold season. In the HW group, a marked decrease in serum glucose and non-esterified fatty acid levels and an increase in insulin and high-density lipoprotein were observed. Additionally, the HW group demonstrated elevated levels of rumen acetate, propionate, isobutyrate, and total volatile fatty acids. Bacterial composition analysis revealed a higher abundance of *Prevotella* and a lower abundance of *NK4A214_group* and *Lachnospiraceae_NK3A20_group* in the RTW group. The expression of metabolites such as Cynaroside A, D-Glucuronic acid, N-acetyl-L-glutamic acid, among others, were found to be significantly downregulated in the RTW group. Additionally, major metabolic pathways, notably nitrogen metabolism, lysine biosynthesis, arginine biosynthesis, and linoleic acid metabolism, were differentially regulated based on water temperature. Our findings emphasize the promise of utilizing heated water as a strategic intervention during colder climates. This approach might serve to optimize energy use and regulate rumen microbial composition and metabolic processes. Further research could focus on the interaction between water temperature and diet composition to improve the performance and health of beef cattle in cold environments.

## Figures and Tables

**Figure 1 metabolites-13-00995-f001:**
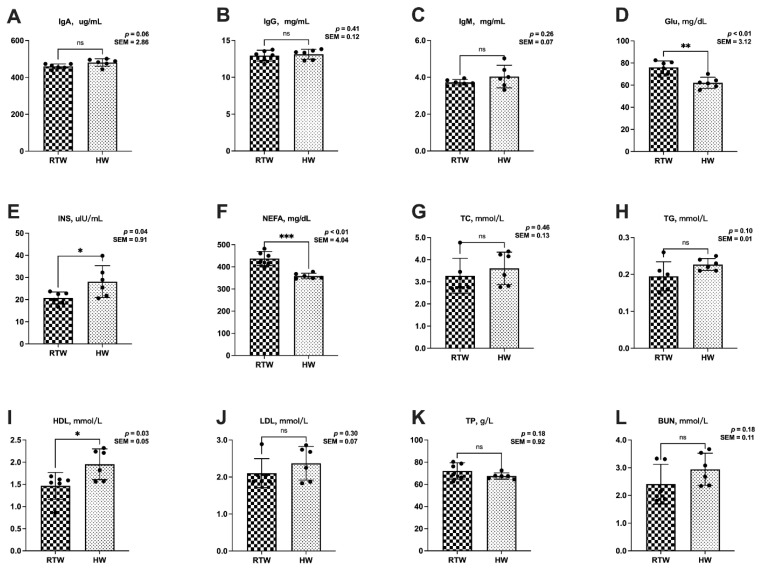
Influence of variations in the water temperature on the serum parameters in beef cattle. (**A**) IgA, immunoglobulin A; (**B**) IgG, immunoglobulin G; (**C**) IgM, immunoglobulin M; (**D**) Glu, glucose; (**E**) INS, insulin; (**F**) NEFA, non-esterified fatty acid; (**G**) TC, total cholesterol; (**H**) TG: triglyceride; (**I**) HDL, high-density lipoprotein; (**J**) LDL, low-density lipoprotein; (**K**) TP, total protein; (**L**) BUN, blood urea nitrogen. RTW and HW correspond to water consumed at 4.39 ± 2.55 °C and water heated to 26.3 ± 1.70 °C, respectively. Different asterisk notations highlight the statistical differences in the results. A single asterisk (*) stands for *p* < 0.05, two asterisks (**) for *p* ≤ 0.01, three asterisks (***) for *p* ≤ 0.001, and (ns) for *p* > 0.05. Sample size, *n* = 6.

**Figure 2 metabolites-13-00995-f002:**
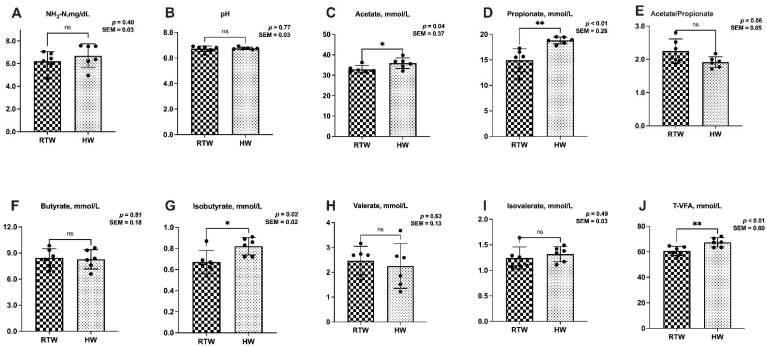
Influence of variations in the water temperature on the rumen fermentation parameters of beef cattle. SEM, standard error of the mean; (**A**) NH_3_-N, ammonia nitrogen; (**B**) pH; (**C**) Acetate; (**D**) Propionate; (**E**) Acetate/Propionate, the ratio of acetate to propionate; (**F**) Butyrate; (**G**) Isobutyrate; (**H**) Valerate; (**I**) Isovalerate; (**J**) T-VFA, total volatile fatty acid; RTW and HW correspond to water consumed at 4.39 ± 2.55 °C and water heated to 26.3 ± 1.70 °C, respectively. Different asterisk notations highlight the statistical differences in the results. A single asterisk (*) signals *p* < 0.05, while double asterisks (**) indicate *p* < 0.01 and (ns) for *p* > 0.05. Sample size, *n* = 6.

**Figure 3 metabolites-13-00995-f003:**
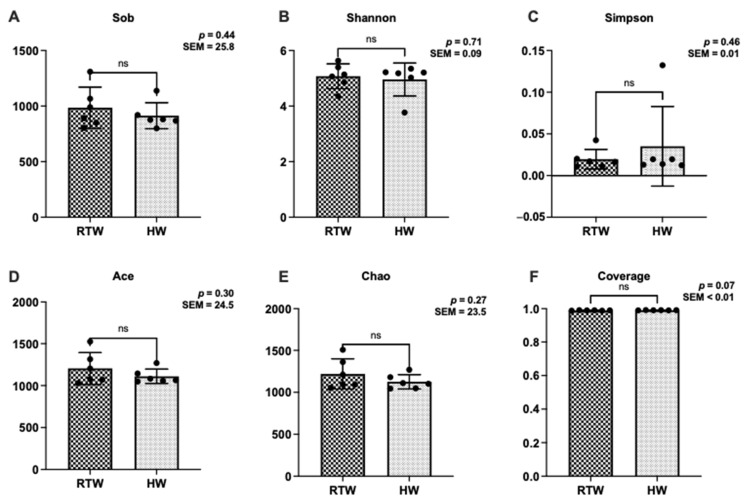
Influence of variations in the water temperature on the bacterial α-diversity of rumen at OTU level in beef cattle. (**A**) Sobs index, species observed index; (**B**) Shannon, Shannon–Wiener diversity index; (**C**) Simpson, Simpson diversity index; (**D**) Ace, abundance-based coverage estimator; (**E**) Chao, Chao1 richness estimator; (**F**) Coverage index, good’s coverage index. RTW and HW correspond to water consumed at 4.39 ± 2.55 °C and water heated to 26.3 ± 1.70 °C, respectively. Sample size, *n* = 6.

**Figure 4 metabolites-13-00995-f004:**
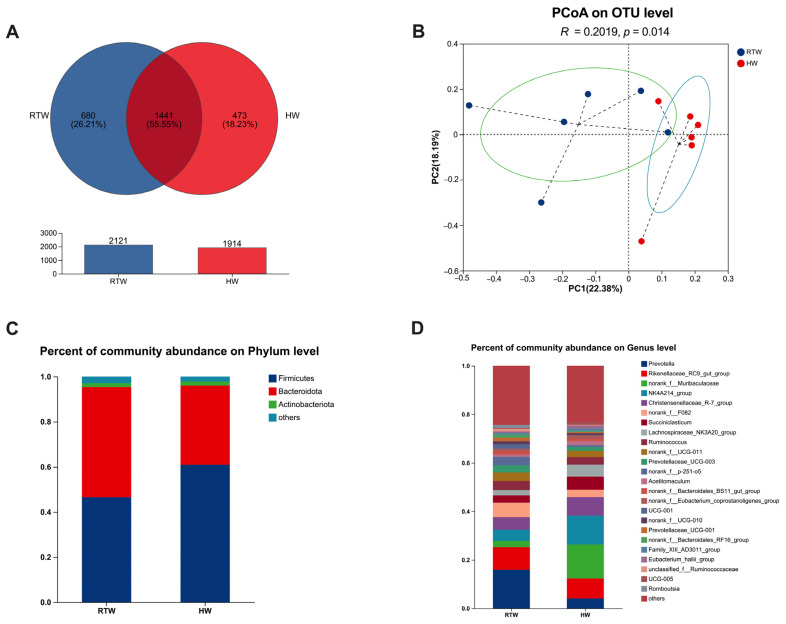
Influence of variations in the water temperature on the rumen bacterial makeup and β-diversity in beef cattle. (**A**) Venn analysis at OUT; (**B**) principal co-ordinate analysis (PCoA) at OTU level; (**C**,**D**) bacterial composition at phylum and genus levels. RTW and HW correspond to water consumed at 4.39 ± 2.55 °C and water heated to 26.3 ± 1.70 °C, respectively. Sample size, *n* = 6.

**Figure 5 metabolites-13-00995-f005:**
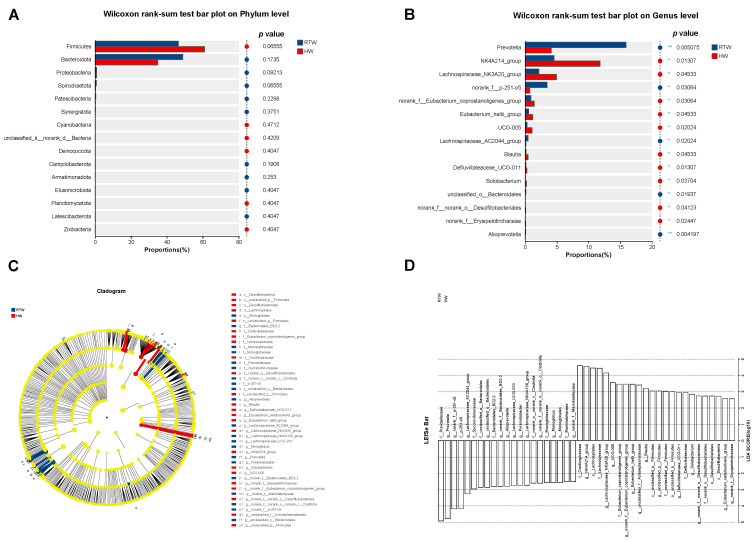
Influence of variations in the water temperature on rumen bacterial composition differences in beef cattle. (**A**,**B**) Variability in microbes at the phylum and genus tiers. (**C**) Phylogenetic tree representation; (**D**) discriminant analysis; LEfSe represents the effect size in linear discriminant analysis. Criteria for representation were *p* < 0.05 and a discriminant analysis score exceeding 2.50. RTW and HW correspond to water consumed at 4.39 ± 2.55 °C and water heated to 26.3 ± 1.70 °C, respectively. Different asterisk notations highlight the statistical differences in the results. A single asterisk (*) signals *p* < 0.05, while double asterisks (**) indicate *p* < 0.01. Sample size, *n* = 6.

**Figure 6 metabolites-13-00995-f006:**
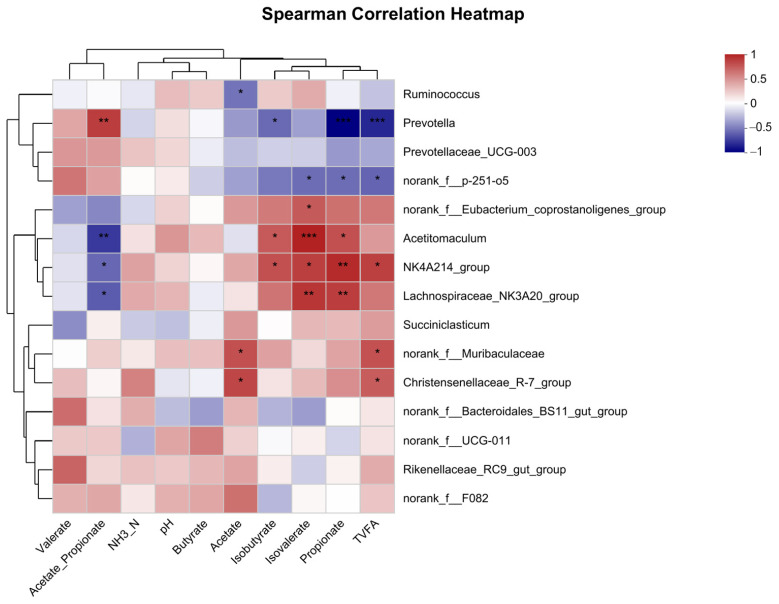
Spearman’s correlation was employed to investigate the associations between the top 15 most prevalent bacterial genera and various parameters. The horizontal axis represents environmental variables, while the vertical axis denotes species. Calculations provide correlation R values and significance levels. Different shades in the visualization represent varying R values, with the legend on the side specifying the range for each color. Acetate_propionate signifies the ratio of acetate to propionate, while TVFAs refers to the total volatile fatty acids. A single asterisk (*) stands for *p* < 0.05, two asterisks (**) for *p* ≤ 0.01, and three asterisks (***) for *p* ≤ 0.001. Sample size, *n* = 6.

**Figure 7 metabolites-13-00995-f007:**
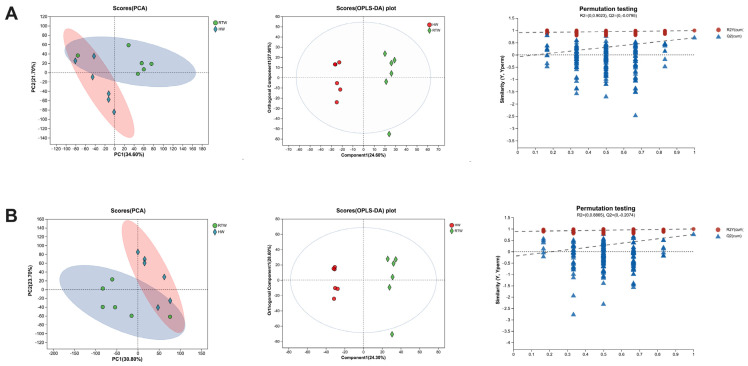
Comparative multivariate statistical evaluations of detected metabolites between the RTW and HW groups. This includes PCA score visualizations, OPLS-DA score graphs, and permutation assessments of the OPLS-DA framework in both positive-ion (**A**) and negative-ion (**B**) modes. Here, PCA denotes the principal component analysis, while OPLS-DA stands for orthogonal projections to latent structures for discriminant analysis. RTW and HW correspond to water consumed at 4.39 ± 2.55 °C and water heated to 26.3 ± 1.70 °C, respectively. Sample size, *n* = 6.

**Figure 8 metabolites-13-00995-f008:**
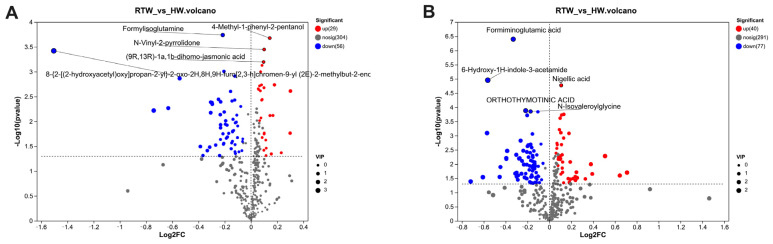
Representation of identified differential metabolites through volcano diagrams in positive-ion (**A**) and negative-ion (**B**) spectrums. In the visuals, red signifies metabolites with heightened concentrations in the HW group, while blue indicates reduced levels compared to the RTW group. Sample size: *n* = 6.

**Figure 9 metabolites-13-00995-f009:**
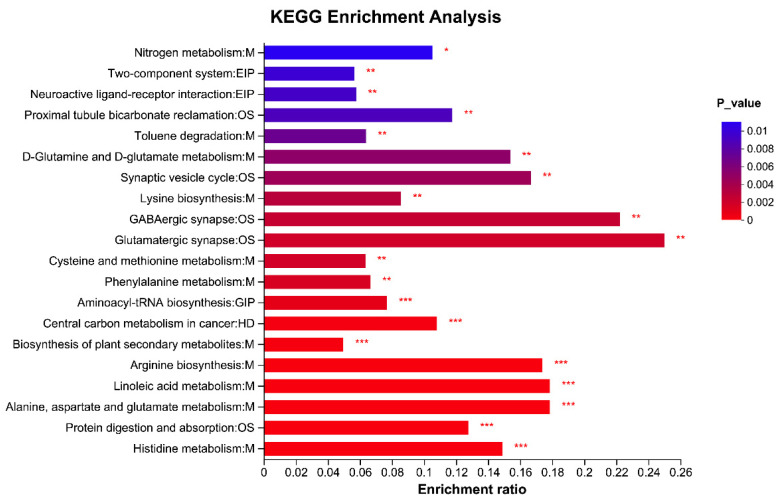
Analysis of the metabolic pathways in rumen fluid from both RTW and HW groups is represented through pathway enrichment. The horizontal axis showcases the specific metabolic pathways, whereas the vertical axis illustrates the degree of enrichment. The varying shades of the bars depict the significance of each pathway’s enrichment, with deeper blue shades signifying more substantial enrichment within the KEGG annotations. Abbreviations in the KEGG annotation include: M for metabolism, EIP for environmental information processing, OS for organismal systems, GIP for genetic information processing, and HD for human diseases. The presence of asterisks on the bars indicates varying levels of significance. A single asterisk (*) stands for *p* < 0.05, two asterisks (**) stand for *p* ≤ 0.01, and three asterisks (***) stand for *p* ≤ 0.001. Sample size, *n* = 6.

**Figure 10 metabolites-13-00995-f010:**
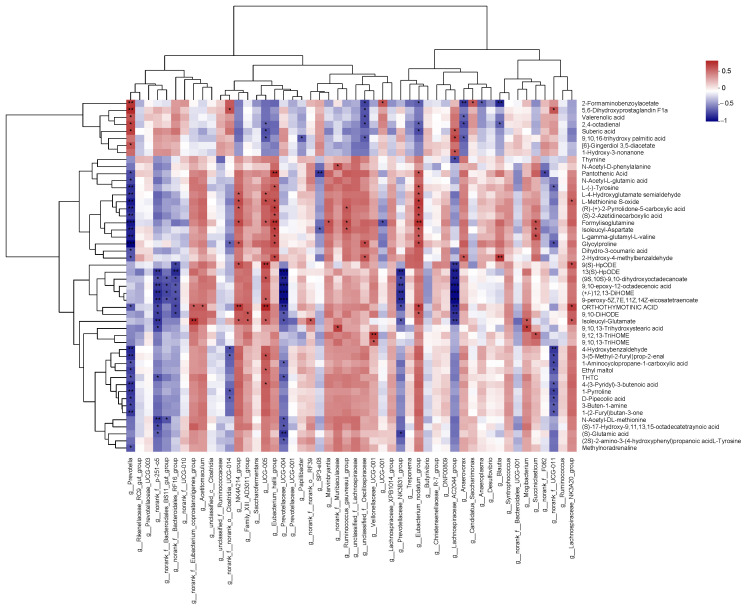
Relationship assessment between genera and metabolite levels influenced by water temperature variations. In the chart, individual rows depict distinct genera, whereas columns signify different metabolites. Each cell within the matrix illustrates the Pearson coefficient, indicating the relationship between a specific genus and metabolite. Positive associations are denoted in red, while negative ones are shown in blue. A single asterisk (*) stands for *p* < 0.05, two asterisks (**) stand for *p* ≤ 0.01, and three asterisks (***) stand for *p* ≤ 0.001. Sample size, *n* = 6.

**Table 1 metabolites-13-00995-t001:** Diet composition and nutrition levels (%, DM basis).

Ingredient Composition	Content
Corn	35.0
Dried distillers grains with solubles	5.00
Corn germ meal	15.0
Cotton seed	4.00
Whole corn silage	34.0
Wheat straw	5.50
Premix	0.50
NaHCO_3_	0.50
NaCl	0.50
Analyzed nutritional composition	
DM	51.2
CP	11.3
ADF	17.4
NDF	32.8
EE	4.07
Calculated nutritional composition	
NEg; Mcal/kg	1.56

DM: dry matter; CP: crude protein; NDF: neutral detergent fiber; ADF: acid detergent fiber; EE: ether extract; premix: Fe 12 g/kg, Mn 1 g/kg, Cu 1 g/kg, Zn 11 g/kg, I 30 mg/kg, Se 30 mg/kg, Co 20 mg/kg, vitamin A 450,000 IU/kg, vitamin D_3_ 60,000 IU/kg, vitamin E 2000 mg/kg; NEg, net energy for growth.

**Table 2 metabolites-13-00995-t002:** Sample sequencing information.

Sample	Sequence Number	Base Number	Mean Length	Min Length	Max Length
RTW_1	54,197	22,604,988	417	215	441
RTW_2	56,093	23,516,338	419	232	504
RTW_3	67,667	28,288,223	418	233	494
RTW_4	63,282	26,418,679	417	201	464
RTW_5	52,643	21,776,008	414	210	512
RTW_6	87,241	36,239,015	415	336	439
HW_1	52,705	21,896,598	415	214	501
HW_2	62,729	25,970,252	414	233	471
HW_3	55,324	23,083,306	417	214	522
HW_4	56,106	23,247,328	414	226	491
HW_5	42,575	17,504,053	411	200	535
HW_6	49,661	20,678,505	416	202	497
Mean	58,352	24,268,608	416	226	489

RTW and HW correspond to water consumed at 4.39 ± 2.55 °C and water heated to 26.3 ± 1.70 °C, respectively.

**Table 3 metabolites-13-00995-t003:** Primary distinct metabolites influencing varied metabolic routes observed between the RTW and HW groups.

Metabolite	VIP	Fold Change	*p* Value	RTW vs. HW
1-Aminocyclopropane-1-carboxylic acid	1.51	0.91	0.02	Down
2-Formaminobenzoylacetate	1.15	1.06	0.02	Up
3-Methyloxindole	1.60	0.87	0.03	Down
4-Methyl-1-phenyl-2-pentanol	1.55	1.10	<0.01	Up
5,6-Dihydroxyprostaglandin F1a	1.04	1.05	<0.01	Up
6-Amino-9H-purine-9-propanoic acid	1.36	1.08	0.03	Up
6-Hydroxy-1H-indole-3-acetamide	2.69	0.68	<0.01	Down
7-ketodeoxycholic acid	1.64	1.19	0.02	Up
Carboxyibuprofen	1.04	1.05	0.01	Up
Cynaroside A	1.96	0.84	<0.01	Down
D-Glucurone	1.18	0.92	<0.01	Down
D-Glucuronic acid	1.39	0.90	0.02	Down
D-Pipecolic acid	2.28	0.81	0.01	Down
Formiminoglutamic acid	2.36	0.80	<0.01	Down
Formylisoglutamine	2.06	0.86	<0.01	Down
Glutamic acid	1.74	0.87	0.03	Up
Hydantoin-5-propionic acid	1.78	0.82	0.03	Down
Indole-3-propionic acid	1.43	0.89	<0.01	Down
Linoelaidic Acid	1.24	0.92	0.04	Down
Melibiitol	1.89	0.81	<0.01	Down
N-Acetyl-L-glutamic acid	1.45	0.91	<0.01	Down
N-Vinyl-2-pyrrolidone	1.27	1.07	<0.01	Up
N-acetyl-L-glutamate-5-semialdehyde	1.04	0.93	0.02	Down
Nigellic acid	1.26	1.08	<0.01	Up
Orthothymotinic acid	2.15	0.86	<0.01	Down
Pantothenic Acid	0.97	0.96	0.02	Down
Tyrosine	1.91	0.84	0.02	Up

The fold change was calculated by dividing the normalized averages of metabolites from the RTW group by those from the HW group. VIP stands for the variable’s significance in the projection.

## Data Availability

The original manuscript of this study is included in the article and further information and data is available upon reasonable request to the corresponding author.

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
