# Peer review of "Effects of Heated Drinking Water during the Cold Season on Serum Biochemistry, Ruminal Fermentation, Bacterial Community, and Metabolome of Beef Cattle"

_metabolites, 2023, doi:10.3390/metabo13090995_

Round 1

Reviewer 1 Report

  1. It appears that the water at room temperature is "cold water," while the heated water is "room temperature water." Kindly provide clarification and make the necessary adjustments.
  2. It is essential to draw distinct conclusions in both the abstract and the main text. The results section currently presents extensive data without offering any evident conclusions.
  3. The titles of each bar in Figure 1 require correction.
  4. In Figure 3, please provide clearer definitions for the abbreviations, such as Chao index, Ace, Simpson, and Coverage index. This will enhance understanding for readers.

Minor English adjustment

Reviewer 2 Report

Comments and Suggestions for Authors

Manuscript ID: metabolites-2573639-peer-review-v1-R1,

entitled " Effects of Heated Drinking Water during Cold Season on Serum Biochemistry, Ruminal Fermentation, Bacterial Community, and Metabolome of Beef Cattle."

General comments.

The manuscript shows that the cold conditions can lower body temperature, affecting performance and immunity. Providing heated drinking water is proposed to mitigate these effects and improve rumen temperature and microbial function. Consequently, heated water enhances cattle growth, antioxidant capacity, and stress resilience.

Research shows water temperature's importance, but mechanisms are unclear. Modern techniques like gene sequencing and mass spectrometry offer insights into rumen microbiota and metabolites.

This study examines how water temperature in cold seasons affects beef cattle's serum biochemistry, rumen microbiota, and metabolites. The goal is to understand how heated water improves health and rumen fermentation, informing livestock feeding strategies in colder climates.

 I accepted this manuscript.

L2 add article the before cold season 

L20 add the full name of OUT.

L20, 26 change Compared to HW to "compared to RTW" and change the result according to this adjustment in the whole manuscript. Because the RTW is considered the control group, the comparison relies on the control group, not vice versa.  

L90 how was drinking water temperature maintained constant at 26.3 ± 1.70°C?

L92 add reference.

 L107 NEg isn’t abbreviated to Metabolic energy but for net energy for growth.

L134 rephrase gauged with measured.

L409 Prevotella make it in italic.

L444-460 check the distance before [references].

In Figure 1 it contains some desirable symbols. It is marked with a red line under it.

Minor editing of English language required

Reviewer 3 Report

The manuscript is not about a novel idea, but is closely related to the conditions of raising beef cows. Instead, it is comprehensive in terms of the research methods used, which allowed the authors to draw correct conclusions.
It requires only minor corrections:
Abstract - L37.38 - please delete the summary on the effect of heated water on stress, as stress parameters were not measured in the manuscript
Material and Methods - please describe the conditions of cold season for the experimental cows.

Round 2

Reviewer 1 Report

The authors response to all reviewer's comments. This manuscript should be considered to publish in current version.